# How Effective is Your Rebuttal? Identifying Causal Models from the OpenReview System

## Abstract

The peer review process is central to scientific publishing, with the rebuttal phase offering authors a critical opportunity to address reviewer concerns. However, the causal mechanisms that determine rebuttal effectiveness, particularly how author responses influence final review decisions, remain poorly understood. In this work, we present a two-layer causal analysis of ICLR submissions from the OpenReview system. At the structured level, we combine metadata features (e.g., soundness, presentation) with LLM-inferred features (e.g., clarity, directness), and apply independence tests to identify their associations with rating changes. At the unstructured level, we model rebuttal texts using a weakly supervised Causal Representation Learning (CRL) framework, guided by LLM-inferred features as concept-level supervision. Theoretically, we establish identifiability conditions for recovering latent concepts under mild assumptions. Empirically, our analysis uncovers causal patterns across structured and unstructured features, revealing how specific rebuttal strategies shape reviewer assessments. These findings offer actionable guidance for authors in crafting more effective rebuttals and contribute to broader goals of transparency, fairness, and efficiency in peer review processes.

## 1 Introduction

Peer review is the cornerstone of scientific progress, ensuring rigor, reliability, and integrity in published research (Tennant, 2018; Alberts et al., 2008; Tennant & Ross-Hellauer, 2020; Ceci & Peters, 1982). In recent years, the machine learning community has advanced transparency by adopting platforms like OpenReview, where reviews, author responses (rebuttals), and ratings are openly available (Tran et al., 2020; Wang et al., 2023a; Sun et al., 2025a; Ross-Hellauer, 2017). This openness presents a rare opportunity to examine how rebuttals influence reviewer judgement, a process often considered important yet poorly understood. Despite its significance, we still lack systematic evidence on what makes rebuttals persuasive and what strategies impact reviewers' ratings.

Prior work on peer review has investigated systemic properties such as bias (Tomkins et al., 2017), arbitrariness (Langford & Guzdial, 2015), predictive validity (Ragone et al., 2013; Wolfram et al., 2020), and integrity (Barrière et al., 2023). Analyses of OpenReview data have studied reviewer behavior and rating consistency (Stelmakh et al., 2021; Gao et al., 2019), while controlled trials examined anchoring effects in scoring (Liu et al., 2024). Several studies have examined the role of rebuttals in the review process. For instance, Shah et al. (2018) found limited score shifts after rebuttal despite active discussion, while Huang et al. (2023) analyzed ICLR 2022 reviews and identified social interaction structures and author strategies that contribute to successful rebuttals. In parallel, Wu et al. (2022) incorporated rebuttal counter-arguments into meta-review generation. However, existing research remains correlational and descriptive, leaving open the causal question of why and how some rebuttal strategies succeed. Please refer to App. A3 for more details about the related work.

Building on the limitations of prior correlational studies, we analyze 8684 ICLR 2024–2025 submissions from the OpenReview system to uncover the causal factors driving score changes. Our analysis consists of two progressive layers. First, we analyze structured features. We construct 24 tabular features, including 8 review-level features (e.g., soundness, presentation), 6 paper-level features (e.g., abstract length), and 10 higher-level LLM-inferred features, which we also refer to as *concepts* (e.g., clarity, directness). These concepts are designed to capture human-aligned signals of rebuttal effectiveness, though they may be noisy or imperfect. To assess their relationship with rating changes,

Table 1: **Dataset statistics of ICLR 2024 and 2025 with paired ratings (initial and final) from Paper Copilot Yang (2025).** The table reports paper-level statistics and review-level statistics.

| | Paper Statistics | | | | | | Review Statistics | | |
|---|---|---|---|---|---|---|---|---|---|
| | Poster | Spotlight | Oral | Reject | Withdraw | Total | Reviewed Papers | Avg. Reviews/Paper | Total Reviews |
| **ICLR'24** | 1321 | 290 | 61 | 1287 | 127 | 3086 | 3086 | 3.9 | 12035 |
| **ICLR'25** | 2412 | 306 | 172 | 2409 | 252 | 5598 | 5598 | 4.1 | 22951 |
| **Total** | 3733 | 596 | 233 | 3696 | 379 | 8684 | 8684 | 4.0 | 35033 |

we apply five conditional independence tests: KCI (Zhang et al., 2012), RCIT (Strobl et al., 2019), HSIC (Gretton et al., 2005), Chi-square (Tallarida et al., 1987), and G-square (Tsamardinos et al., 2006). Second, we extend the analysis with textual causal representation learning (CRL). Rather than relying only on predefined, noisy LLM-inferred *concepts*, we aim to directly recover latent *concepts* from rebuttal and review text. This second layer serves two purposes. First, it corrects for the noise and potential biases in the LLM-inferred *concept* by learning representations that more closely reflect the underlying causal factors. Second, it adds new *concepts* beyond our predefined set, uncovering dimensions of rebuttal effectiveness that may not be anticipated by human intuition. Recent advances in CRL enable the identification of interpretable *concepts* from high-dimensional text (Schölkopf et al., 2021; Yao et al., 2024), especially when supported by weak supervision with *concept* cues (Rajendran et al., 2024; Locatello et al., 2020) and variation induced by multiple distributions or interventions (Zhang et al., 2024; Ahuja et al., 2023). In this way, CRL builds directly on the structured analysis, refining noisy *concept* features while also discovering new latent *concepts*, thereby providing a richer understanding of how rebuttal strategies shape reviewer assessments.

To summarize, our contributions are twofold. First, we conduct a deep and wide-ranging analysis of rebuttal effectiveness across structured features, covering paper-level metadata, review-level features, and LLM-inferred *concepts*. By applying established independence tests in this setting, we provide a comprehensive examination and extract insightful patterns about which factors are dependent from rating changes. Second, we frame rebuttal effectiveness as a causal modeling problem and situate it within a multi-distribution framework. Building on this formulation, we introduce a new causal model and establish novel identifiability results for recovering latent *concepts* from rebuttal and review text. These two layers are progressive: the structured layer offers weak supervision to guide CRL, while CRL both refines noisy concepts and uncovers additional ones beyond the predefined set. Together, these contributions advance the theoretical foundations and empirical understanding of rebuttals, while offering practical guidance for crafting more effective responses in OpenReview.

## 2 DATASETS

**Data Collection and Processing.** We build on data from Paper Copilot (Yang, 2025), a website launched two years ago to aggregate and analyze AI conference data. Since 2024, Paper Copilot has compiled peer review records from major conferences, and for ICLR it provides both initial and final reviewer ratings. We focus on ICLR 2024 and 2025 submissions because they not only include these paired ratings, which are crucial for analyzing rating changes, but also offer the most complete author–reviewer discussion records available in the OpenReview system. In contrast, other conferences in Paper Copilot often release only final ratings, making it impossible to study rating changes. Our dataset includes reviews, rebuttals, and subsequent discussions. In total, we collect 8696 papers, with additional statistics reported in Tab. 1. To ensure fair and meaningful analysis, we filter out papers without rebuttals. After processing, we obtain 23922 valid reviewer–author discussion samples, each with paired ratings. This dataset provides sufficient scale for both independent test and CRL, enabling us to study rebuttal effectiveness from complementary perspectives.

**Metadata Features.** Each paper is typically evaluated by multiple reviewers, so we collect 8 review-level features (*initial confidence, final confidence, confidence change, number of interactions, soundness, presentation, contribution, and the average of other reviewers' initial ratings*) and 6 paper-level features (*submission number, title length, abstract length, number of authors, status, and primary area*). Here, the number of interactions denotes how many rounds of discussion occur between the author and a reviewer. The feature average of other reviewers' initial ratings is designed to capture how peer assessments may influence an individual reviewer. Together, these metadata

Table 2: **Comparison of LLMs in scoring concept features.** We report the $L_2$ norm between LLM predictions and human annotations for 10 concept features: *Clarity (CL), Directness (DI), Attitude (AT), Author Openness (AO), Evidence (EV), Rigor (RI), De-Escalation (DE), Specificity and Constructiveness (SC), Reviewer Openness (RO), and Concern Severity (CS)*. The final column shows the average $L_2$ error *(AE)* across all features, and *TC* denotes the average time cost.

| Models (LLMs) | Metrics | | | | | | | | | | |
|---|---|---|---|---|---|---|---|---|---|---|---|
| | TC↓ | CL↓ | DI↓ | AT↓ | AO↓ | EV↓ | RI↓ | RI↓ | SC↓ | RO↓ | CS↓ | AE↓ |
| DeepSeek-R1 | 18.33s | 0.30 | 0.55 | 0.47 | 0.60 | 0.83 | 0.63 | 0.67 | 0.80 | 0.55 | 0.80 | 0.62 |
| Grok-3-Latest | 11.74s | 0.45 | 0.50 | 0.74 | 0.60 | 0.39 | 0.53 | 0.67 | 1.00 | 0.90 | 1.30 | 0.71 |
| Gemini-2.0-Flash-Lite | 3.70s | 0.40 | 0.75 | 0.53 | 0.95 | 0.89 | 0.58 | 1.00 | 0.30 | 1.10 | 0.65 | 0.71 |
| ChatGPT-4.1-Mini | 9.73s | 0.35 | 0.65 | 0.95 | 0.95 | 0.94 | 0.63 | 1.00 | 1.25 | 0.60 | 0.65 | 0.80 |
| Gemini-2.0-Flash | 4.12s | 0.45 | 1.10 | 0.79 | 0.95 | 1.11 | 0.74 | 0.83 | 0.75 | 1.05 | 0.45 | 0.82 |
| ChatGPT-4.1 | 9.73s | 0.55 | 0.70 | 0.79 | 1.05 | 0.89 | 0.84 | 0.67 | 1.40 | 0.90 | 0.65 | 0.84 |
| ChatGPT-4.1-Nano | 5.34s | 0.35 | 0.55 | 1.05 | 1.30 | 0.72 | 0.58 | 2.50 | 0.50 | 0.50 | 0.75 | 0.88 |
| Llama-4-Maverick | 5.86s | 0.50 | 0.75 | 1.26 | 1.35 | 1.28 | 0.89 | 2.00 | 0.75 | 0.70 | 0.75 | 1.02 |
| Gemini-2.5-Flash-Preview-04-17 | 13.26s | 0.65 | 0.75 | 1.35 | 1.35 | 0.94 | 0.84 | 1.17 | 1.45 | 1.20 | 1.25 | 1.03 |
| Deepseek-V3-0324 | 5.94s | 0.55 | 1.00 | 1.84 | 0.80 | 1.28 | 1.16 | 1.50 | 1.40 | 1.30 | 0.85 | 1.17 |
| ChatGPT-4o-Latest | 8.43s | 0.60 | 1.30 | 2.00 | 1.40 | 1.17 | 0.95 | 2.17 | 2.00 | 1.05 | 0.80 | 1.34 |

features reflect both the content and dynamics of the peer review process. The App. A4.2 and App. A2 present a full description of all features and their distributions respectively. Together, these features serve as the foundation for our structured analysis.

**LLM-Inferred Features.** To capture finer-grained aspects of rebuttals and reviews, we introduce ten human-interpretable features, hereafter referred to as *concepts*. Seven are author-related: *clarity*, *directness*, *attitude*, *openness*, *evidence*, *rigor*, and *de-escalation*. Three are reviewer-related: *specificity and constructiveness*, *openness*, and *concern severity*. Formal definitions and empirical distributions of these features are provided in App. A4.1 and App. A1. All features are annotated on a five-point ordinal scale. To scale annotation beyond manual labeling, we benchmarked several LLMs on a randomly sampled set of 200 annotated examples (20 rebuttals evaluated across 10 features). Each example was independently labeled by two senior machine learning researchers, with disagreements resolved through discussion to produce gold-standard scores. We then compared LLM-inferred predictions against expert annotations using the $L_2$ norm.

The results in Tab. 2 show that `DeepSeek-R1` achieves the best overall alignment with human annotations, with the lowest average $L_2$ error across all concept features, albeit at a rather costly inference time. `Grok-3-Latest` and `Gemini-2.0-Flash-Lite` perform competitively (average error 0.71) while being faster, but with higher variance across specific features. In contrast, models such as `ChatGPT-4o-Latest` and `DeepSeek-V3-0324` exhibit significantly larger errors ($>1.0$), indicating weaker consistency with expert labels. Overall, `DeepSeek-R1` providing the best performance for large-scale inference. Based on this result, we adopt `DeepSeek-R1` for large-scale inference. Due to budget and inference time constraints, we annotate a 10% subset of the dataset (2393 samples) using `DeepSeek-R1`. The full prompt is provided in Prompt 1 and 2.

## 3 FIRST LAYER: STRUCTURED FEATURE ANALYSIS

To identify which factors influence reviewer ratings, we investigate how review- and rebuttal-related features are associated with rating changes. For this purpose, we apply five widely used independence tests: KCI (Zhang et al., 2012), RCIT (Strobl et al., 2019), HSIC (Gretton et al., 2005), Chi-square (Tallarida et al., 1987), and G-square (Tsamardinos et al., 2006). The tests are conducted on two sets of features: (i) 14 metadata features available for all 23922 reviewer–author discussions, and (ii) the annotated subset (2393 samples) with 10 LLM-inferred features. At the review level, Fig. 1 reports the aggregated independence test results between ratings and metadata or LLM-inferred features. At the paper level, Fig. 2 shows the aggregated results between the average paper rating and the six paper-level metadata features. Together, these analyses provide a comprehensive view of how different types of features may be linked to changes in reviewer ratings.

**Analysis of review-level metadata features.** As shown in Fig. 1 (orange), for both initial and final ratings, we find strong dependence on core review attributes such as *soundness*, *presentation*, and

Figure 1: **Aggregated independence test results** between rating and metadata features (orange) or LLM-inferred features (blue). Each cell reports how many of the five tests fail to reject independence at significance level $\alpha{=}0.05$. A value of 0 indicates strong evidence of dependence, while 5 indicates strong evidence of independence across all tests. Refer to Fig. A5.2-A5 for all complete $p$ values.

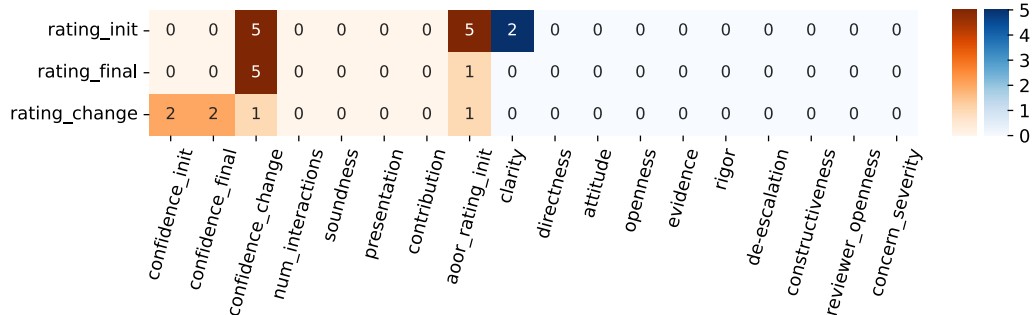

*contribution*. This pattern is consistent across all five tests, indicating that these factors reliably shape how reviewers assign scores. Interestingly, *rating_init* and *rating_final* appear independent of each other but both are dependent on *rating_change*, suggesting that absolute ratings and their shifts capture complementary aspects of the review process. Rating changes show weaker but notable dependence on *initial/final confidence* and on the *average of other reviewers' initial ratings*, highlighting the role of confidence and peer influence in shaping score adjustments. Finally, the *number of interactions* also exhibits dependence with *rating_change*, underscoring the importance of active reviewer–author engagement during the discussion phase.

**Analysis of LLM-inferred features.** Also in Fig. 1 (blue), turning to the ten concept features extracted from rebuttal and review text, we observe systematic dependence with rating changes. Features such as *clarity*, *directness*, *attitude*, *openness*, *evidence quality*, and *rigor* all show strong associations with score adjustments, suggesting that the style and substance of rebuttals are key drivers of reviewer updates. Unlike review-level metadata, which primarily governs baseline ratings, these content-oriented concepts appear to capture how authors' responses shift reviewers' perceptions. This supports the intuition that rebuttal effectiveness is closely tied to the persuasiveness and tone of the exchange.

Figure 2: **Aggregated independence test results** between average rating and paper-level metadata features.

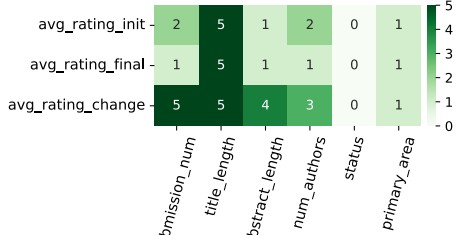

**Analysis of paper-level metadata features.** As shown in Fig. 2, static paper descriptors such as *title length*, *abstract length*, *number of authors*, and *primary area* exhibit little evidence of association with rating changes. In most cases, the tests fail to reject independence, particularly for average rating adjustments, indicating that these surface-level attributes have minimal impact on how scores evolve during the rebuttal stage. Weak dependencies are observed for *submission number*, *abstract length*, and *number of authors* with initial ratings, but these effects vanish once rating changes are considered. This suggests that while such metadata may weakly shape first impressions, it does not determine whether reviewers subsequently adjust their scores. Taken together, these findings underscore an important distinction: paper-level descriptors capture static characteristics of the submission, but rebuttal effectiveness hinges on dynamic interactions between authors and reviewers. In other words, what changes minds is not how long the title is or how many co-authors are listed, but rather the persuasiveness, clarity, and responsiveness demonstrated during the rebuttal process.

**Takeaway.** Our findings have several insights for the machine learning community. We highlight that rebuttal effectiveness depends more on discourse quality than on paper metadata. For authors, features such as *clarity*, *rigor*, *evidence*, and *constructiveness* are most associated with rating gains, whereas soundness, presentation, contribution, and confidence measures show little effect (see Fig.3). For reviewers and area chairs, this underscores the need for calibration, as specific rebuttal strategies can systematically shape score adjustments. We also observe a regression-to-the-mean

Figure 3: **Dependency panels of rating changes and various features.** Each subplot shows the distribution of rating_change conditioned on metadata or LLM-inferred features. Most features are centered around zero, indicating little systematic effect. Notable dependencies appear for *aoor_rating_init* (lower initial ratings linked to positive changes, consistent with a regression-to-the-mean effect), *num_interactions* (more exchanges associated with slight gains), and discourse-related features such as *clarity*, *rigor*, *evidence*, and *constructiveness*, which show modest positive shifts. Appendix A6 provide the corresponding dependency panels for *rating_init* and *rating_final*.

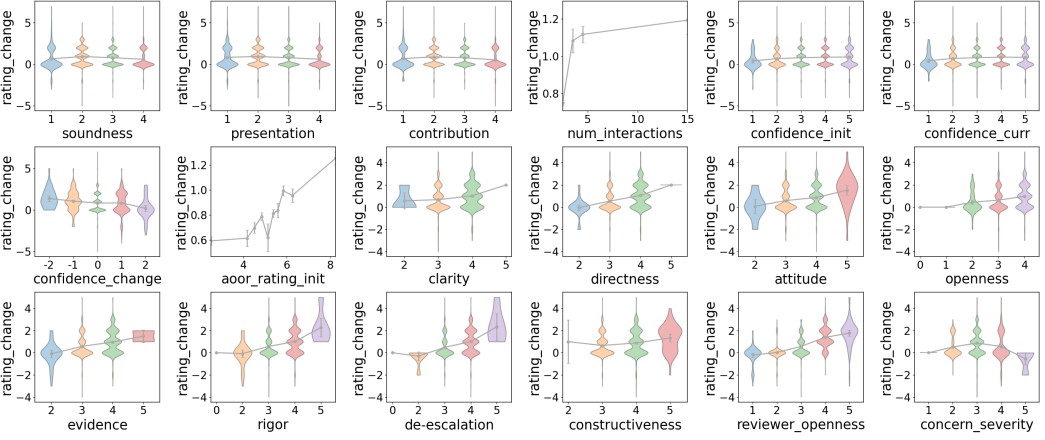

effect, where lower initial ratings tend to increase, and a modest positive trend with more reviewer–author interactions. Overall, the substance and quality of rebuttals matter in influencing final rating.

# 4 SECOND LAYER: HUMAN-ALIGNED CAUSAL REPRESENTATION LEARNING

Beyond structured metadata features, rebuttals involve high-dimensional textual interactions between reviewers and authors. To move beyond predefined and potentially noisy features, we employ causal representation learning (CRL) (Xu et al., 2024; Zheng et al., 2022; Yao et al., 2023; Sun et al., 2025b). These two layers are progressive: the structured feature analysis in the previous section provides weak supervision for CRL, while CRL serves two complementary functions. First, it refines noisy LLM-inferred concepts by recovering latent representations that better align with the true underlying concepts. Second, it uncovers additional human-aligned concepts beyond the predefined set, capturing aspects of rebuttal effectiveness that may not be immediately apparent. Our setting naturally exhibits multiple modalities (reviewer text and author text) and heterogeneous distributions (e.g., different primary areas, reviewer backgrounds), which provide the variation necessary for causal identification. Fig. 4 illustrates that the causal model of the rebuttal process with two modalities. We organize this section as follows. In

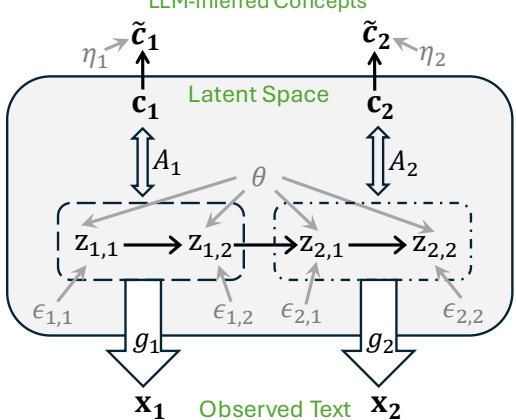

Figure 4: **Causal model of the rebuttal process with two modalities**, capturing latent factors from reviewers ($z_1$) and authors ($z_2$). The framework allows bidirectional causal influences between $z_1$ and $z_2$, reflecting the interactive nature of rebuttal discussions. $x$ denotes the observed text, $c$ the true human-aligned concepts, and $\tilde{c}$ the noisy estimation inferred by LLMs used as weak supervision.

§ 4.1, we introduce our causal formulation, motivated by the real-world rebuttal problem. § 4.2 develops the identifiability theory grounded in this formulation. Building on these insights, § 4.3 presents our network design and training procedure. To evaluate the effectiveness of the proposed method, we conduct synthetic experiments in § 4.4. Finally, § 4.5 applies our approach to the rebuttal dataset, providing the actionable insights and practical guidance on effective rebuttal strategies.

## 4.1 CAUSAL FORMULATION

We denote the observed text as $\mathbf{x} = [\mathbf{x}_1, \mathbf{x}_2]$, where $\mathbf{x}_1$ represents aggregated reviewer text and $\mathbf{x}_2$ aggregated author text. These are generated from latent variables $\mathbf{z} = [\mathbf{z}_1, \mathbf{z}_2]$, which capture hidden factors underlying how reviewers and authors express themselves, respectively. We introduce $\theta$ to represent the heterogeneity of latent variables across different primary areas or reviewer backgrounds.

On top of the latent variables, we define a set of human-aligned concepts $\mathbf{c} = [\mathbf{c}_1, \mathbf{c}_2]$, which correspond to interpretable attributes such as *openness*. Each concept is modeled as a linear projection of $\mathbf{z}$ through a mapping $A$. In practice, however, LLMs-inferred scores only provide noisy approximations $\tilde{\mathbf{c}}$ of these concepts, which we generally represent this relation as $\tilde{c}_{m,i} = c_{m,i} + \eta_{m,i}$, where $\eta$ are Gaussian noises introduced by annotation variability, prompt ambiguity, or model bias.

Formally, the data-generating process of the variables defined above can be written as:

$$z_{m,i} := h_{m,i}(\text{Pa}(z_{m,i}), \theta, \epsilon_{m,i}), \quad \textit{(latent causal relations)} \tag{1}$$

$$\mathbf{x}_m := g_m(\mathbf{z}_m), \quad \textit{(observed text)} \tag{2}$$

$$\mathbf{c}_m := A\,\mathbf{z}_m, \quad \textit{(true human-aligned concepts)} \tag{3}$$

$$\tilde{c}_{m,i} := c_{m,i} + \eta_{m,i}, \quad \textit{(noisy concepts inferred by LLMs)} \tag{4}$$

where $m \in \{1, 2\}$ indexes reviewers ($m{=}1$) and authors ($m{=}2$). $\text{Pa}(z_{m,i})$ are the parents of $z_{m,i}$ in the latent causal graph $\mathcal{G}_{\mathbf{z}}$, $\epsilon_{m,i}$ are exogenous noise variables, $g_m$ is a nonlinear mixing function mapping $\mathbf{z}_m$ to observed text $\mathbf{x}_m$, and $A$ is a linear matrix aligning $\mathbf{z}_m$ with human-interpretable concepts $\mathbf{c}_m$. Notably, we treat the noisy LLM-derived $\tilde{\mathbf{c}}$ as weak supervision for CRL training; ultimately, CRL refines them to recover the true concept $\mathbf{c}$ by utilizing the estimated $\hat{\mathbf{z}}$, thereby offering a more reliable concept representation beyond LLM-inferred results.

## 4.2 IDENTIFIABILITY THEORY

A central question in our setting is whether human-aligned concepts $\mathbf{c}$ can be uniquely recovered from the observational data. More concretely, we are given an observational dataset (all reviews) together with multiple concept-conditional datasets (subsets of reviews filtered by concept scores) with LLM-inferred noisy concept; the fundamental problem is to determine the conditions under which the true underlying concepts can be recovered from them with minor indeterminacy.

**Definition 1.** *(**Identifiability**) Given observational and concept-conditional datasets, we say the concepts $\mathbf{c} = \{c^1, \ldots, c^m\}$ with associated linear maps $\{A^1, \ldots, A^m\}$ are **identifiable** if, for any alternative parameterization $(\tilde{f}, \tilde{A}^e, \tilde{b}^e)$ that produces the same observed data distributions, there exists an invertible linear map $T$, a shift $w \in \mathbb{R}^{d_{\mathbf{z}}}$, permutation matrices $P^e$, and invertible diagonal matrices $\Lambda^e$ such that, for all $\mathbf{x}$ and for each concept $e$, the concept parameters are related by*

$$\tilde{A}^e \tilde{f}^{-1}(\mathbf{x}) = \Lambda^e P^e A^e (f^{-1}(\mathbf{x}) + w), \quad \tilde{A}^e = P^e A^e T^{-1}, \quad \tilde{b}^e = \Lambda^e P^e (b^e - A^e w). \tag{5}$$

In this context, *identifiability* means that the human-aligned concepts $\mathbf{c}$ can be recovered up to a small set of unavoidable ambiguities. Specifically, in Eq. 5, the subspaces corresponding to interpretable dimensions such as *soundness* or *clarity* and their evaluation maps can be consistently learned, modulo permutation ($P^e$), scaling ($\Lambda^e$), and a global linear transformation of the latent space ($A$). These ambiguities are intrinsic to causal representation learning, since the latent variables $\mathbf{z}$ are never directly observed; however, they do not obstruct our objective if we can recover them up to minor indeterminacy. Learning the evaluation maps $A^e f^{-1}$ allows us to dissect rebuttals, identify which latent factors causally drive reviewer perceptions, and align the results with interpretable axes such as rigor, evidence, or openness. Importantly, the conditions for identifiability, relying on the diversity of concept-conditional distributions, are naturally satisfied in our setting due to the rich variety of sub-score data available in the OpenReview System.

**Theorem 1.** *(Identifiability of Concepts) Suppose we match the observations $\mathbf{x}_m$ across modalities (authors and reviewers), and the following conditions hold in the data-generating process:*

   *i* *(Information Preservation): The functions $g_1$ and $g_2$ are differentiable and invertible.*

   *ii* *(Sufficient Diversity): All entries of $v^\top B$ are non-zero, where $B_{i,j} = \frac{b_k^e}{\sigma^2}$ denotes the area–concept matrix.*

   *iii* *(Distinctive Concept Alignment): There exists a set of linearly independent aligning vectors $\mathcal{C} = \{a_1, \ldots, a_n\}$ such that, for each concept $C^e$, the rows of the aligning matrix $A^e$ lie in $\mathcal{C}$, i.e., $(A^e)^\top e_i \in \mathcal{C}$. Let $S^e$ denote the indices of the subset of $\mathcal{C}$ that appear as rows of $A^e$. Every aligning vector in $\mathcal{C}$ appears in at least one primary area $e$ (where an area corresponds to a concept-conditional distribution), that is,*

$$\bigcup_e S^e = [n].$$

*Then the concepts $\mathbf{c}$ are identifiable as demonstrated in Definition 1.*

Assumption i requires that the latent space is recoverable from the observed data. Assumption ii further requires the presence of latent distribution shifts in the review concepts across different primary areas, ensuring variability in the underlying structure. Finally, Assumption iii ensures that all concepts can be decomposed into a finite set of atomic components that remain distinct across primary areas, which is essential. Under such a theorem, we can guarantee that the hidden concepts are uniquely recovered and aligned with the LLM-inferred noisy version, ensuring the correctness of the obtained causal relations on those latent embeddings through conditional independence testing.

### 4.3 NETWORK DESIGN

Building on the identifiability conditions, we now describe our practical framework for learning latent causal representations from rebuttal and review text. The key challenge is to approximate the posterior distribution of latent variables $\mathbf{z}$ given observed embeddings $\mathbf{x}$, while accounting for domain-specific variation $\theta$ (e.g., primary research areas or reviewer backgrounds). Since the generative process $\mathbf{x} = g(\mathbf{z})$ is highly nonlinear, the posterior $p(\mathbf{z}|\mathbf{x}, \theta)$ is intractable. We therefore adopt a variational approach enhanced with flow-based priors.

Following the general CRL framework (Zhang et al., 2024), we use a Dense Sigmoid Flow (DDSF) (Wehenkel & Louppe, 2019) to implement the prior distribution over $\mathbf{z}$. The flow transforms each dependent latent variable $z_i$ into an independent noise variable $\epsilon_i$, conditioned on its latent parents and domain factor $\theta$. Let $W$ denote the adjacency matrix of the latent causal graph $\mathcal{G}_{\mathbf{z}}$, and $\hat{W}$ be the estimated one. For each $\hat{z}_i$, the transformation is:

$$\hat{\epsilon}_i, \ \log \det J_i = \text{Flow}\Big(\hat{z}_i; \ \text{NN}(\{\hat{W}_{i,j} \ \hat{z}_j\}_{j<i}, \theta)\Big), \tag{6}$$

where $\hat{\epsilon}_i$ is the transformed independent noise, $\log|J_i|$ is the Jacobian determinant, and NN denotes a neural network generating flow parameters. Assuming $\epsilon$ is factorized (e.g., $\mathcal{N}(0, I)$), the prior distribution is:

$$\log p(\hat{\mathbf{z}}; \theta) = \sum_i \Big( \log p(\hat{\epsilon}_i) + \log |J_i| \Big). \tag{7}$$

We implement an encoder $q_\phi(\hat{\mathbf{z}}|\mathbf{x}, \theta)$ that maps observed text embeddings $\mathbf{x}$ into an approximate posterior over latent variables $\hat{\mathbf{z}}$. A decoder $p_\psi(\mathbf{x}|\mathbf{z}, \theta)$ reconstructs the observed embeddings from the true latent variables $\mathbf{z}$. The model is trained by maximizing the domain-conditioned Evidence Lower Bound (ELBO):

$$\mathcal{L}_{\text{ELBO}} = \mathbb{E}_{q_\phi(\hat{\mathbf{z}}|\mathbf{x}, \theta)}[\log p_\psi(\mathbf{x}|\hat{\mathbf{z}}, \theta)] - D_{\text{KL}}\big(q_\phi(\hat{\mathbf{z}}|\mathbf{x}, \theta) \,\|\, p(\mathbf{z}; \theta)\big), \tag{8}$$

where $\phi$ are encoder parameters, $\psi$ are decoder parameters, and $p(\mathbf{z}; \theta)$ is the flow-based prior. To ensure the learned space is both causally structured and human-aligned, we add two regularizers:

$$\mathcal{L}_{\text{sparsity}} = \|\hat{W}\|_1, \quad \mathcal{L}_{\text{supervision}} = \frac{1}{K} \sum_{k=1}^{K} \big(\hat{\tilde{c}}_k - \tilde{c}_k\big)^2, \tag{9}$$

where $\hat{\tilde{\mathbf{c}}}$ are the estimated noisy concepts, $\tilde{\mathbf{c}}$ are LLM-inferred concepts, and $K$ is the total latents.

Table 3: **Synthetic experiment results across different configurations.** We evaluate our method against four baselines: $\beta$-VAE (reconstruction loss only), iVAE (independent VAE), Sun et al. (multi-modal only), and Zhang et al. (multi-domain only). Results are reported as Pearson MCC and Spearman MCC percentages. Our method consistently achieves higher MCC across most configurations, demonstrating the effectiveness of combining multi-modal and multi-domain information.

| Method | Linear | | | | Nonlinear | | | |
|---|---|---|---|---|---|---|---|---|
| | Gaussian | | Laplacian | | Gaussian | | Laplacian | |
| | Pearson | Spearman | Pearson | Spearman | Pearson | Spearman | Pearson | Spearman |
| **Y Structure** | | | | | | | | |
| $\beta$-VAE | 73.00±5.2 | 73.80±4.8 | 70.77±6.1 | 70.31±5.9 | 86.11±3.2 | 88.64±2.9 | 75.34±4.7 | 75.96±4.3 |
| iVAE | 51.63±8.4 | 51.12±7.9 | 39.33±9.2 | 37.00±8.7 | 53.02±6.8 | 51.41±6.5 | 12.25±3.1 | 13.43±2.8 |
| Sun et al. | 82.86±4.1 | 82.70±3.8 | 72.59±5.3 | 73.89±4.9 | 69.49±6.2 | 70.86±5.7 | 61.60±7.4 | 71.16±6.8 |
| Zhang et al. | 70.32±6.8 | 70.30±6.2 | 63.60±8.1 | 66.57±7.5 | 72.22±5.9 | 72.66±5.4 | 48.33±9.2 | 50.70±8.7 |
| **Ours** | **84.33±6.17** | **86.08±6.02** | **81.01±7.31** | **83.38±9.16** | **82.79±2.73** | **83.65±3.33** | **71.38 ± 2.82** | **77.58 ± 3.98** |
| **Chain Structure** | | | | | | | | |
| $\beta$-VAE | 77.95±4.8 | 79.66±4.2 | 77.14±5.1 | 77.30±4.6 | 72.08±6.3 | 74.04±5.8 | 69.79±7.2 | 71.33±6.9 |
| iVAE | 46.99±7.2 | 45.83±6.8 | 46.47±8.1 | 49.04±7.6 | 50.87±5.9 | 49.32±5.4 | 8.87±2.4 | 12.44±2.1 |
| Sun et al. | 86.52±3.7 | 87.06±3.4 | 68.13±6.8 | 70.28±6.2 | 65.43±7.9 | 65.61±7.3 | 59.63±8.6 | 70.47±7.8 |
| Zhang et al. | 70.55±6.4 | 69.63±5.9 | 62.24±7.8 | 66.16±7.2 | 69.90±5.6 | 70.93±5.1 | 46.16±8.9 | 48.86±8.4 |
| **Ours** | **80.33±11.29** | **81.00±11.87** | **81.88±6.71** | **83.72±7.13** | **79.60±5.53** | **80.99±5.75** | **68.87 ± 2.51** | **76.78 ± 2.34** |

**Final objective.** The overall objective is:

$$\mathcal{L}_{\text{total}} = \mathcal{L}_{\text{ELBO}} + \lambda_1 \mathcal{L}_{\text{sparsity}} + \lambda_2 \mathcal{L}_{\text{supervision}}, \tag{10}$$

with $\lambda_1, \lambda_2$ balancing reconstruction, causal sparsity, and concept alignment. This design ensures that the model (i) captures domain-dependent causal mechanisms via the flexible flow-based prior, (ii) recovers human-interpretable dimensions by refining noisy LLM-derived concepts $\tilde{\mathbf{c}}$ into true latent concepts $\mathbf{c}$, and (iii) retains additional capacity to uncover novel concepts beyond the predefined set.

## 4.4 SYNTHETIC EXPERIMENTS

**Baselines and Metrics.** We evaluate our method against four representative CRL baselines, each emphasizing different modeling assumptions and trade-offs. $\beta$-VAE (Higgins et al., 2017) relies only on reconstruction loss and encourages disentanglement but lacks explicit identifiability guarantees. iVAE (Tomczak & Welling, 2018) introduces identifiable priors but does not exploit multi-domain or multi-modal variation effectively. Sun et al. (Sun et al., 2025b) leverage multi-modal information but assume a single domain setting, while Zhang et al. (Zhang et al., 2024) leverage multi-domain variation in a general setting but are limited to a single modality. Our approach integrates both multi-domain and multi-modal information simultaneously, directly addressing the limitations of these baselines. For evaluation metrics, we adopt both Pearson MCC and Spearman MCC. Pearson measures linear alignment between learned and ground-truth causal factors, while Spearman assesses monotonic rank-order consistency across values. Using both together provides a robust and comprehensive view of causal discovery performance, with higher values indicating better recovery.

**Implementation Details.** We generate synthetic data with four latent variables under two canonical causal structures: Y-structure and Chain-structure. Each domain contains 10,000 samples, with 20 heterogeneous simulated domains in total. The model architecture consists of an encoder–decoder with two hidden layers (64 and 32 units), trained for 1,000 iterations using Adam with learning rate 0.001 throughout. Key hyperparameters are: reconstruction weight 5.0, KL divergence weight 0.1, sparsity weight 0.01, and supervision weight 1.0. We test both Gaussian and Laplacian priors extensively, with evaluation every 50 iterations, averaged across multiple random seeds for robustness.

**Results and Analysis.** Table 3 summarizes results across all configurations. First, iVAE consistently performs the weakest, especially under nonlinear settings (e.g., Spearman MCC below 15% in nonlinear Laplacian cases), showing that identifiable VAEs alone are insufficient without domain or modality variation. Second, $\beta$-VAE performs reasonably well in linear settings (above 70%), but performance degrades significantly under nonlinear distributions, reflecting its lack of identifiability.

Third, Sun et al. (multi-modal only) and Zhang et al. (multi-domain only) improve over $\beta$-VAE and iVAE in certain cases, but both struggle when only one type of variation is available. For instance, Sun et al. underperform in nonlinear chain structures, while Zhang et al. show sharp drops under nonlinear Laplacian settings. Finally, our method consistently achieves the best or near-best MCC across all configurations, improving by 3–10 points over the strongest baseline. Notably, in challenging nonlinear Laplacian settings, our approach maintains high correlations (e.g., 77.58% Spearman in Y-structure and 76.78% in Chain-structure), while baselines deteriorate. These results confirm that leveraging both multi-domain and multi-modal information is critical for robust causal representation learning. Our method not only recovers known causal factors more faithfully but also demonstrates strong generalization across different domains.

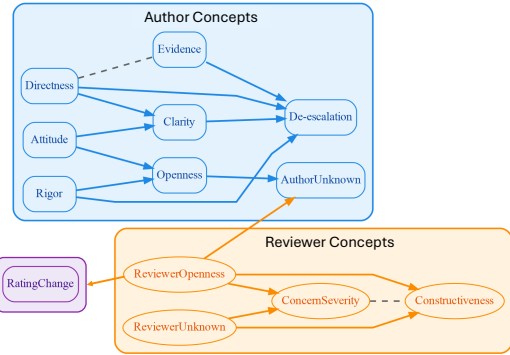

Figure 5: **Real-world experiment result.** This causal graph is learned by our proposed CRL method. Excepts the 10 given concepts, we also learn additional two concepts. See § 4.5 for details.

## 4.5 REAL-WORLD EXPERIMENT ON ICLR REBUTTAL DATASET

We apply our CRL framework to the ICLR rebuttal dataset to learn latent variables **z** from review and rebuttal text, and then use causal discovery methods to recover the underlying causal graph (Fig. 4). Using 2393 labeled samples with 10 LLM-inferred concept labels, plus extensive unlabeled data, we train a MultiModalMarkovFlowVAE with separate encoders for reviewer (4 latent variables) and author (8 latent variables) modalities, each discovering one additional unknown concept. The learned causal graph (Fig. 5) shows that on the author side, *Clarity* plays a central role, influencing *Directness*, *Attitude*, and *De-escalation*, while *Openness* connects to *Rigor* and *Evidence*, highlighting how clear, transparent responses shape both tone and substance. On the reviewer side, *Review Quality*, *Reviewer Openness*, and *Concern Severity* form a tightly connected cluster, with *Reviewer Openness* directly driving *Rating Change*. Notably, two emergent latent concepts enrich this picture: the *Author_Unknown* node, connected to openness and influencing reviewer openness, likely reflects hidden aspects of persuasiveness or tone in author responses, while the *Reviewer_Unknown* node, linked with review quality and concern severity and directly affecting rating change, appears to capture latent reviewer dispositions such as strictness or flexibility. These results show that rebuttal effectiveness arises from both author strategies (clarity, evidence, rigor) and reviewer disposition, while latent factors further reveal subtle but impactful influences beyond predefined features.

## 5 DISCUSSIONS AND CONCLUSION

**Discussions.** A key limitation of our study is that the analysis is restricted to reviews from the ICLR 2024 and 2025, which may limit the generalizability of the findings to other conferences or time periods. In addition, due to data availability constraints, we aggregate each paper's reviews without accounting for the precise timestamps of individual revisions. As a result, our analysis do not capture the temporal dynamics of how rebuttals and reviewer ratings evolve over the review process.

**Conclusions.** We presented a two-layer causal analysis of rebuttal effectiveness in ICLR 2024–2025 submissions. At the structured layer, independence tests on metadata and LLM-inferred *concepts* revealed that clarity, directness, rigor, and evidence are most strongly linked to rating changes, while static paper descriptors play little role. At the unstructured layer, our causal representation learning framework refined noisy LLM-derived *concepts* and uncovered new latent dimensions, supported by identifiability guarantees. Together, these findings provide both theoretical insights into causal modeling of text and actionable guidance for the ML community: authors can focus on substantive, evidence-based rebuttals, while reviewers and chairs should remain aware of systematic influences on scoring. Our work thus contributes toward a more transparent, fair, and effective peer review process.

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

*Appendix for*

## "How Effective is Your Rebuttal? Identifying Causal Models from the OpenReview System"

Table of Contents:

## A1 ETHICS STATEMENT

All datasets used in this work are restricted to non-commercial, academic research purposes. We obtained the necessary permissions from the respective platforms. A summary of the applicable terms-of-use is as follows:

- **ICLR** – Submissions and reviews are hosted on OpenReview under an open-access license (CC BY 4.0), which explicitly permits reuse for research purposes.
- **PaperCopilot** – According to its Terms of Use, no part of the Site's content may be copied, reproduced, distributed, or otherwise exploited for commercial purposes without express prior written permission. We obtained such permission and consent directly from the website to enable non-commercial and academic research use.

## A2 THE USE OF LARGE LANGUAGE MODELS (LLMs)

We employ large language models (LLMs) to infer author- and reviewer-related concepts from textual inputs in our dataset. To ensure robustness and model fit, we conducted a comparative evaluation across candidate LLMs and selected the most suitable model for this task (Tab. 2). In line with community guidance on LLM usage, we explicitly disclose this use and retain full responsibility for the accuracy and integrity of all LLM-derived outputs. For transparency and reproducibility, we also provide the exact inference prompt used in our pipeline (App. 1), enabling independent verification and replication of our procedure.

## A3 DETAILS ABOUT RELATED WORK

### A3.1 PEER REVIEW ANALYSIS

Peer review in scientific publishing has been widely studied, with work addressing bias (Tomkins et al., 2017), consistency (Langford & Guzdial, 2015), and predictive validity (Ragone et al., 2013). The transparency of the OpenReview platform has further enabled analyses of reviewer behavior and decision-making (Stelmakh et al., 2021; Gao et al., 2019). Recent studies provide complementary perspectives. (Liu et al., 2024) conducted a randomized controlled trial and found that reviewers are not strongly anchored to their initial scores, showing a willingness to revise after rebuttals, though the drivers of such changes remain unclear. The LazyReview dataset (Purkayastha et al., 2025) addresses a different challenge by identifying low-effort or vague reviews, offering tools to improve review quality. By contrast, the effectiveness of rebuttals themselves has received relatively limited attention. (Shah et al., 2018) showed that rebuttals lead to score changes in about 25% of reviews, while (Gao et al., 2019) explored correlates of successful rebuttals without establishing causality. Our work extends these efforts by explicitly modeling the causal mechanisms underlying rebuttal effectiveness.

### A3.2 CAUSAL REPRESENTATION LEARNING

Causal representation learning (CRL) seeks to uncover latent causal factors from high-dimensional data (Schölkopf et al., 2021; Parascandolo et al., 2018), enabling reasoning about interventions and counterfactuals. Recent work has shown that CRL can learn disentangled representations capturing causal mechanisms (Lachapelle et al., 2022; Lippe et al., 2022), making it particularly useful in domains where causal factors are latent or noisy, such as peer review. Unsupervised CRL methods face identifiability challenges (Locatello et al., 2019), which researchers have attempted to address using temporal structure (Klindt et al., 2020), sparsity assumptions (Bengio et al., 2019), or group-theoretic frameworks (Besserve et al., 2018). However, such assumptions often fail in real-world settings. To overcome this, weak supervision and multi-environment data have been proposed to improve identifiability (Locatello et al., 2020; Shu et al., 2020). Building on weakly supervised approaches (Shen et al., 2022) and concept-based representation learning (Rajendran et al., 2024), our work adapts these ideas to model rebuttal effectiveness.

### A3.3 NATURAL LANGUAGE PROCESSING FOR SCIENTIFIC TEXT

Analyzing rebuttals requires handling complex scientific text. Advances in natural language processing have enabled richer analysis of scientific documents (Beltagy et al., 2019; Cohan et al., 2020), supporting tasks such as classification, summarization, citation intent detection (Cohan et al., 2019; Jurgens et al., 2018), document retrieval (Wang et al., 2023b), and fact-checking (Wadden et al., 2022). While less explored, rebuttals have been studied through argument mining (Lawrence & Reed, 2020; Fromm et al., 2021) and persuasive language (Tan et al., 2016), reflecting their persuasive nature in influencing reviewer opinions.

Our work connects these directions by applying causal representation learning to study rebuttal effectiveness in scientific peer review, focusing on the OpenReview system in machine learning conferences.

## A4 DETAILS ABOUT THE DATASET AND ANALYSIS

### A4.1 EXPLANATION OF CONCEPT FEATURES IN TAB.2

We consider ten variables capturing key aspects of rebuttals and reviews. *Clarity (CL)* reflects how clearly the rebuttal communicates its arguments, while *Directness (DI)* measures the extent to which it addresses reviewer concerns explicitly. *Attitude (AT)* captures the tone of the rebuttal, distinguishing professional and respectful responses from defensive ones. *Authors Openness (AO)* denotes the willingness of authors to acknowledge limitations or alternative perspectives. *Evidence (EV)* refers to the use of data, experiments, or citations to support claims, and *Rigor (RI)* evaluates the technical soundness and thoroughness of rebuttal arguments. *De-Escalation (DE)* reflects the ability to resolve misunderstandings and reduce conflict during the exchange. On the reviewer side, *Review Quality*

*(RQ)* measures the specificity and constructiveness of feedback, *Reviewer Openness (RO)* captures the willingness of reviewers to revise their evaluation in light of rebuttals, and *Concern Severity (CS)* indicates the seriousness of the issues raised in the review.

### A4.2 EXPLANATION OF VARIABLES IN FIG.1

We further include metadata and reviewer-provided variables. *Title Length* and *Abstract Length* measure the verbosity of the submission's title and abstract, respectively, while *Num Authors* captures the number of contributing authors. *Status* indicates the acceptance outcome (e.g., oral, poster, reject), and *Primary Area* records the main research domain of the paper. *Num Interactions* reflects the extent of back-and-forth exchanges between authors and reviewers. Reviewer scores are also considered: *Soundness* assesses methodological correctness, *Presentation* evaluates clarity of exposition, and *Contribution* reflects novelty and significance. Finally, *aoor_rating_diff* measures the average of other reviews's rating differences or changes for one reviewer; we define this variable in order to see how one reviewer can be influenced by other reviewers.

### A4.3 DATASET ANALYSIS

For fine-grained analysis, we annotate a 10% random sample of the dataset with interpretable labels capturing both rebuttal quality and reviewer behavior. Rebuttal-related dimensions include *Clarity, Directness in Addressing Reviewer Concerns, Positive Attitude, Willingness to Acknowledge Limitations, Strength of Evidence, Technical Convincingness and Rigor, and Handling of Misunderstandings and De-escalation*, while reviewer-related dimensions include *Review Specificity and Constructiveness, Open-mindedness, and Severity of Concerns*. All labels are rated on a 5-point ordinal scale, with detailed guidelines provided in the annotation prompt (Appendix).

To construct the annotated subset, we manually labeled 20 review–rebuttal threads and used them to benchmark 10 LLMs. We then computed the L-2 distance between model predictions and human labels across dimensions. As shown in Table 2, DeepSeek-R1 achieved the closest alignment to human annotations and was chosen to label the full 10% set.

In addition to the annotated labels, we extract further labels from OpenReview, including metadata such as *Title Length, Abstract Length, Number of Authors, Status, Primary Area, and Number of Reviewer-Author(s) Interactions*, as well as reviewer-provided scores *Soundness, Presentation, Contribution, Initial Rating, Final Rating, Initial Confidence, and Final Confidence*. Using this subset, we conduct pairwise independence tests with Kernel-based Conditional Independence (KCI), Randomized Conditional Independence (RCSI), Hilbert-Schmidt Independence Criterion (HSIC), Chi-squared, and G-squared tests. Detailed results for each method are given in the Appendix A5.2. Figure 1 summarizes the findings, where each cell shows how many of the five tests failed to reject the null hypothesis.

The aggregated results in Figure 1 reveal several patterns. *Rating Difference* shows strong dependence with *Openness*, *Evidence*, and *Rigor*, suggesting that reviewers who initially gave low scores are more likely to revise them when faced with open, well-supported, and rigorous rebuttals. *Number of Interaction* is also dependent on *Rating Difference*, reflecting the role of back-and-forth communication in driving score changes. By contrast, *Clarity*, *Directness*, and *Attitude* show no dependence with *Rating Difference*, likely due to their skewed distribution (most rebuttals score highly, leaving little variability) or the selection bias of top-tier conference submissions, where both papers and reviews tend to be of consistently high quality. Interestingly, *Clarity* and *Attitude* do show dependence with *Initial Rating* and *Final Rating*, but not with *Rating Difference*, implying that they shape the overall impression of a paper without directly influencing score updates.

We also find that *Reviewer Openness* and *Severity of Concerns* are strongly associated with *Rating Difference*, indicating that large score changes occur when open-minded reviewers engage with rebuttals addressing serious issues. In contrast, metadata features such as *Title Length*, *Abstract Length*, *Number of Authors*, and *Primary Area* show no dependence on *Rating Difference*, suggesting they play only a minor role compared to content-based signals. The dependence of *Status* on *Initial Rating*, *Final Rating*, and *Rating Difference* is expected, as decisions (e.g., oral, poster) follow review scores. Finally, *Confidence* scores appear largely independent of other features, suggesting they are influenced by external factors.

Figure A1: Distributions of features

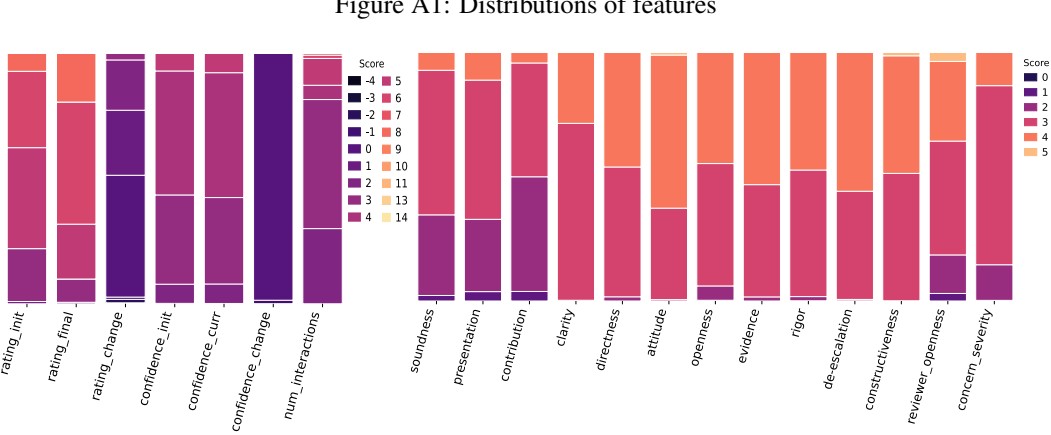

Figure A2: Paper-level metadata feature distributions; see Appendix A1 for primary area details.

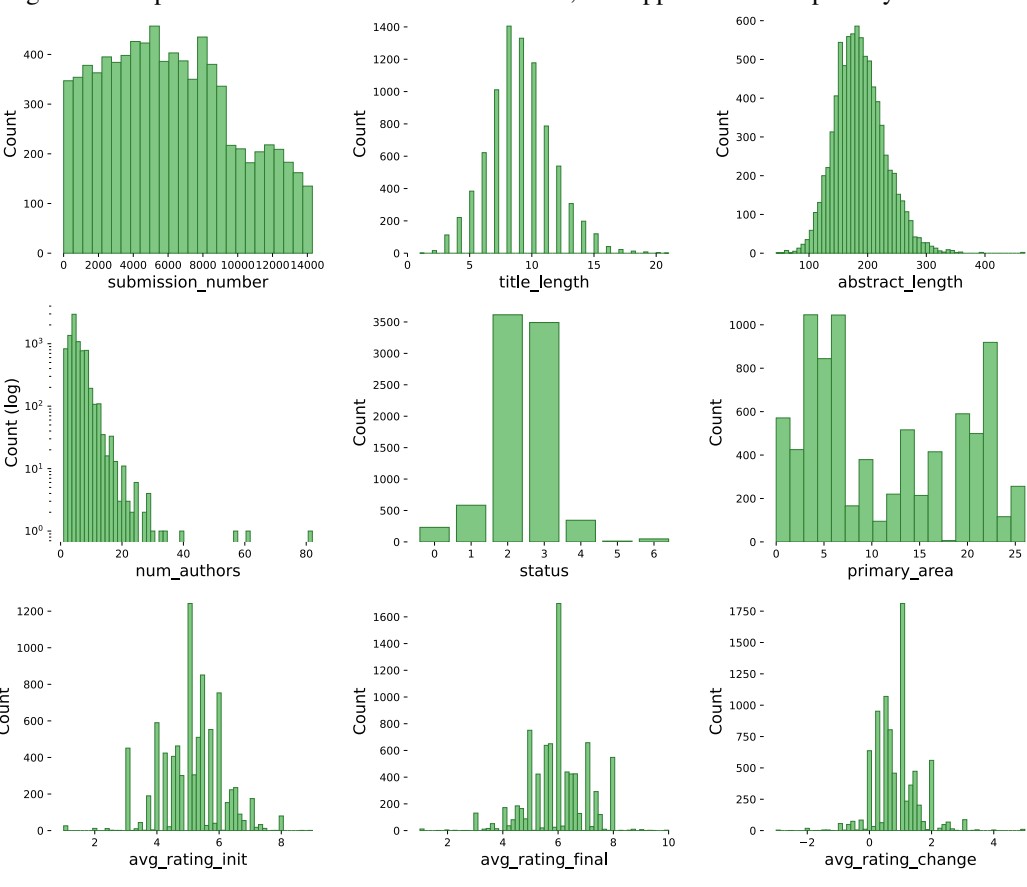

## A5 LEARNING HUMAN-ALIGNED CAUSAL REPRESENTATIONS

### A5.1 BASIC CONCEPT

To connect abstract latent variables with human-understandable criteria, we model review subscores (e.g., soundness, clarity, novelty) as *concepts*. Formally, each concept is defined as a linear projection $A : \mathbb{R}^{d_z} \to \mathbb{R}^{d_C}$ of the latent rebuttal representation $\mathbf{z}$, with a valuation $b \in \mathbb{R}^{d_C}$ corresponding to the reviewer's assigned subscore (e.g., clarity = 4). Thus, rebuttals with the same subscore form

a concept-conditional set in latent space. This formulation anchors the learned representations to interpretable axes aligned with reviewer evaluations.

### A5.2 Identifiability of Causal Models: Theorem and Proof

**Theorem 1.** *(Identifiability of Review Concepts) Suppose we match the observations $\mathbf{x}_m$ across modalities (authors and reviewers), and the following conditions hold in the data-generating process:*

   i  *(Information Preservation): The functions $g_1$ and $g_2$ are differentiable and invertible.*

   ii  *(Primary Area Diversity): All entries of $v^\top B$ are non-zero, where $B_{i,j} = \frac{b_k^e}{\sigma^2}$ denotes the area–concept matrix.*

   iii  *(Thought Reflection): The latent components in $\mathbf{z}_1$ are causal parents of $\mathbf{z}_2$, but not vice versa.*

   iv  *(Distinctive Concept Alignment): There exists a set of linearly independent aligning vectors $\mathcal{C} = \{a_1, \ldots, a_n\}$ such that, for each concept $C^e$, the rows of the aligning matrix $A^e$ lie in $\mathcal{C}$, i.e., $(A^e)^\top e_i \in \mathcal{C}$. Let $S^e$ denote the indices of the subset of $\mathcal{C}$ that appear as rows of $A^e$. Every aligning vector in $\mathcal{C}$ appears in at least one area $e$ (where an area corresponds to a concept-conditional distribution), that is,*

$$\bigcup_e S^e = [n].$$

*Then the review concepts are identifiable as in Definition 1.*

**Discussion of Assumptions** Assumption i requires that the latent space is recoverable from the observed data. Assumption ii further requires the presence of latent distribution shifts in the review concepts across different primary areas, ensuring variability in the underlying structure. Assumption iii reflects the natural process in which authors first read the reviews, then engage in reflection, and finally provide rebuttals. Finally, Assumption iv ensures that all concepts can be decomposed into a finite set of atomic components that remain distinct across primary areas, which is essential for separating and identifying them.

**Proof Sketch** We first recover the latent space from the reviews and author responses by applying the inverse generating functions together with the fixed causal direction between the author and review modules. The presence of latent distribution shifts in the review concepts across different primary areas then provides additional variation, which allows us to identify each concept by comparing the concept spaces across environments. In this way, the atomic concepts can be causally inferred.

**Overview.** We prove that the review-side concepts are identifiable in the OpenReview system under Assumptions i–iv. Authors and reviewers provide two observed modalities $(\mathbf{x}_1, \mathbf{x}_2)$ generated from latent variables $(\mathbf{z}_1, \mathbf{z}_2)$. True human-aligned concepts are linear functionals of the latents, $\mathbf{c} = A\mathbf{z}$, while the LLM only yields noisy surrogates $\tilde{\mathbf{c}}$ defined by $\tilde{c}_{m,i} = c_{m,i} + \eta_{m,i}$ with Gaussian noise $\eta_{m,i}$. Our argument follows five steps: first we pass to a canonical latent parameterization; next we obtain a key observable identity; then we identify the concepts in different primary areas; after that, we recover primary-area valuations; finally, we remove residual symmetries to obtain uniqueness of the concept coordinates (up to permutation and scaling), matching Definition 1. Throughout this proof, "environments" are *primary areas* of OpenReview.

Assumption i states that both observation maps $g_1, g_2$ are differentiable and invertible, hence
$$(\mathbf{z}_1, \mathbf{z}_2) = (g_1^{-1}(\mathbf{x}_1), g_2^{-1}(\mathbf{x}_2)).$$

Assumption iii fixes the causal direction: reviewer latents $\mathbf{z}_2$ are parents of response latents $\mathbf{z}_1$ (reviews influence responses), which rules out label-swap artifacts between the two modalities.

Let the atomic concept directions be the rows of the alignment map $A$, denoted $C = \{a_1, \ldots, a_n\}$. For each primary area $e$, let $A^e$ collect the active rows and $b^e$ be the associated valuations. Introduce the primary-area–concept incidence matrix $M \in \mathbb{R}^{m \times n}$ and the primary-area–valuation matrix $B \in \mathbb{R}^{m \times n}$ by

$$M_{ei} = \begin{cases} \sigma_i^{-2}, & \text{if } a_i \text{ is a row of } A^e, \\ 0, & \text{otherwise,} \end{cases} \qquad B_{ei} = \begin{cases} \sigma_i^{-2} b_k^e, & \text{if the } k\text{-th row of } A^e \text{ equals } a_i, \\ 0, & \text{otherwise.} \end{cases} \tag{11}$$

Writing $q_{\sigma^2}$ for a centered Gaussian with variance $\sigma^2$, the primary-area densities satisfy

$$\ln p(\mathbf{z}) - \ln p_e(\mathbf{z}) = \sum_{i=1}^{n} \left( \frac{1}{2} M_{ei} \langle a_i, \mathbf{z} \rangle^2 - B_{ei} \langle a_i, \mathbf{z} \rangle \right) + c_e, \tag{12}$$

for constants $c_e$.

To place the model in standard coordinates, pick an invertible $T \in \mathbb{R}^{d_z \times d_z}$ with $T^{-\top} a_i = e_i$ for $1 \leq i \leq n$, a shift $\lambda \in \mathbb{R}^{d_z}$ with $\lambda_i = 0$ for $i > n$, and a diagonal matrix $\Sigma$ with $\Sigma_{ii} = \sigma_i$ for $i \leq n$ and $\Sigma_{ii} = 1$ otherwise. Define the affine reparameterization

$$L(z) = \Sigma^{-1} T z - \lambda. \tag{13}$$

Push the model through $L$:

$$z = L(\mathbf{z}), \qquad g_m \leftarrow g_m \circ L^{-1}, \qquad A^e \leftarrow A^e T^{-1}, \qquad p(z) = p(L^{-1}z) \, |\det T^{-1}|. \tag{14}$$

If the $k$-th row of $A^e$ equals $a_i$, update

$$b_k^e \leftarrow b_k^e / \sigma_i - \lambda_i. \tag{15}$$

In this standard form all nonzero entries of $M$ are 1, and

$$M = M \operatorname{Diag}(\sigma_1^2, \ldots, \sigma_n^2), \qquad B = B \operatorname{Diag}(\sigma_1^{-1}, \ldots, \sigma_n^{-1}) - M \operatorname{Diag}(\lambda_1, \ldots, \lambda_n). \tag{16}$$

The observed distributions are unchanged (only the Jacobian modifies densities), so both parameterizations generate the same $(\mathbf{x}_1, \mathbf{x}_2)$. Choose $\lambda$ so each row of $B$ has mean zero across primary areas, and flip any coordinate $z_i$ so that the first nonzero entry in column $i$ of $B$ is positive. Assumption ii is stable under this normalization: $v^\top M = 0$ and $v^\top B \neq 0$ before the transformation implies the same after diagonal rescaling and centering. We henceforth work in this standard form.

Define the latent log-density contrasts

$$g_e(z) := \ln p_0(z) - \ln p_e(z) = \sum_{i=1}^{n} \left( \tfrac{1}{2} M_{ei} z_i^2 - B_{ei} z_i \right) - c_e', \tag{17}$$

where $p_0$ is the observational mixture over primary areas. On the observation side, set

$$G_e(x) := \ln p_X^0(x) - \ln p_X^e(x). \tag{18}$$

Because $g_m$ are diffeomorphisms on the data manifold, Jacobians cancel in differences and

$$g_e(z) = G_e\big(g(z)\big) = G_e(x). \tag{19}$$

Even when $g$ maps into a submanifold (if $d_z \neq d_x$), local charts yield the same difference; hence $G_e$ is identifiable from data and so are geometric features of argmin sets of $\sum_{e \in T} g_e$.

Let $S^e = \{ i \in [n] : a_i \text{ is a row of } A^e \}$ be the active atoms in area $e$. For $T \subset [m]$, write $S^T = \bigcup_{e \in T} S^e$ and consider

$$I_T := \arg\min_z \sum_{e \in T} g_e(z). \tag{20}$$

Since $g_e$ are convex quadratics that separate across coordinates, there exist univariate convex functions $h_i^T$ with

$$\sum_{e \in T} g_e(z) = \sum_{i=1}^{n} h_i^T(z_i), \qquad h_i^T(z_i) = \begin{cases} \text{strictly convex with unique minimizer } z_i^T, & i \in S^T, \\ 0, & i \notin S^T. \end{cases} \tag{21}$$

Therefore

$$I_T = \{ z \in \mathbb{R}^{d_z} : z_i = z_i^T \text{ for all } i \in S^T \}, \qquad \dim(I_T) = d_z - |S^T|. \tag{22}$$

Using $g_e(z) = G_e(g(z))$,

$$g(I_T) = \arg\min_x \sum_{e \in T} G_e(x), \tag{23}$$

and $\dim g(I_T) = \dim(I_T)$ because $g$ is a diffeomorphism on the data manifold. Hence $|S^T|$ is identifiable for every $T \subset [m]$. In particular, $n = |S^{[m]}|$ (each atom appears in at least one primary area by Assumption iv).

Knowing all $|S^T|$ recovers $M$ up to a permutation of columns (relabeling concepts). An induction on $m$ is standard: when $m = 1$, $|S^{\{1\}}|$ counts active atoms in the first area; for the step $m \to m+1$, the values $|S^{T \cup \{m+1\}}|$ across $T \subset [m]$ reveal which columns satisfy $M_{m+1,i} = 1$, and the differences $|S^T| - |S^{T \cup \{m+1\}}|$ identify the columns with $M_{m+1,i} = 0$. Thus $M$ is identified up to column permutation.

Fix an atom $i$ and define

$$T_i = \{e \in [m] : M_{ei} = 0\}. \tag{24}$$

By Assumption iv (distinctive alignment), for every $i' \neq i$ there exists an area that filters $i'$, hence $S^{T_i} = [n] \setminus \{i\}$. Consequently,

$$I_{T_i} = \{z \in \mathbb{R}^{d_z} : z_{i'} = z_{i'}^{T_i} \text{ for all } i' \neq i\}, \tag{25}$$

so only $z_i$ varies on $I_{T_i}$. For any area $e$ with $i \in S^e$ (so $M_{ei} = 1$ in standard form),

$$g_e(z) = c_e^{T_i} + \frac{1}{2} z_i^2 - B_{ei} z_i \qquad \text{on } I_{T_i}. \tag{26}$$

If $e_1 \neq e_2$ both contain $i$, define the slice where $g_{e_1}$ is minimized:

$$I_{T_i}^{e_1} = \arg \min_{z \in I_{T_i}} g_{e_1}(z) = \{z \in I_{T_i} : z_i = B_{e_1 i}\}. \tag{27}$$

Evaluating $g_{e_2}$ on $I_{T_i}^{e_1}$ and subtracting its minimum over $I_{T_i}$ gives

$$\min_{z \in I_{T_i}^{e_1}} g_{e_2}(z) - \min_{z \in I_{T_i}} g_{e_2}(z) = \frac{(B_{e_1 i} - B_{e_2 i})^2}{2}. \tag{28}$$

The left-hand side is observable since $g(I_{T_i}) = \arg \min_x \sum_{e \in T_i} G_e(x)$ is identifiable and we can minimize $G_{e_2}$ over $g(I_{T_i})$. Hence $|B_{e_1 i} - B_{e_2 i}|$ is identified for all pairs with $i$ active. Choose the pair with maximal separation to bracket all $B_{ei}$ for $i \in S^e$; the zero-mean row constraint from Step 1 fixes the additive constant and the "first nonzero positive" convention fixes the sign. Repeating over $i$ identifies $B$ (up to the same column permutation as $M$).

Consider two standard-form representations $(z, g, p)$ and $(\tilde{z}, \tilde{g}, \tilde{p})$ that share $(M, B)$. Let $\varphi = \tilde{g}^{-1} \circ g$. Decompose $z = (z_c, z_o)$ with $z_c \in \mathbb{R}^n$ the concept coordinates and $z_o \in \mathbb{R}^{d_z - n}$ the complement; fix $z_o$ and set $\iota_o(z_c) = (z_c, z_o)$ and $\varphi_o(z_c) = \pi_c(\varphi(\iota_o(z_c)))$, where $\pi_c$ projects to the first $n$ coordinates. Since $g_e$ depends only on $(M, B)$ in both models,

$$g(\iota_o(z_c)) = G(g(\iota_o(z_c))) = G(\tilde{g}(\varphi(\iota_o(z_c)))) = g(\varphi_o(z_c)), \tag{29}$$

with $g = (g_e)_{e=1}^m$ and $G = (G_e)_{e=1}^m$. Differentiating,

$$D_i g_e(z) = M_{ei} z_i - B_{ei}, \qquad Dg(z) = M \operatorname{Diag}(z_1, \ldots, z_n) - B, \tag{30}$$

so for $z = \iota_o(z_c)$ and $\tilde{z} = \iota_o(\varphi_o(z_c))$,

$$M \operatorname{Diag}(z_1, \ldots, z_n) - B = \left( M \operatorname{Diag}(\tilde{z}_1, \ldots, \tilde{z}_n) - B \right) D\varphi_o(z_c). \tag{31}$$

Let $M^+$ be a left pseudoinverse of $M$ (rank $n$ holds by coverage and linear independence), and choose $v$ from Assumption ii with $v^\top M = 0$ and $v^\top B \neq 0$. Stacking yields

$$\tilde{M}^+ = \begin{pmatrix} M^+ \\ v^\top \end{pmatrix} \in \mathbb{R}^{(n+1) \times m}, \tag{32}$$

and multiplying,

$$\begin{pmatrix} z_1 & 0 & \cdots & 0 \\ & \ddots & \ddots & \vdots \\ 0 & \cdots & z_n & 0 \\ 0 & \cdots & 0 & 0 \end{pmatrix} - \tilde{M}^+ B = \left( \begin{pmatrix} \tilde{z}_1 & 0 & \cdots & 0 \\ & \ddots & \ddots & \vdots \\ 0 & \cdots & \tilde{z}_n & 0 \\ 0 & \cdots & 0 & 0 \end{pmatrix} - \tilde{M}^+ B \right) D\varphi_o(z_c). \tag{33}$$

Table A1: Primary areas and counts

| ID | Primary Area | Counts |
|---|---|---|
| 0 | General machine learning (i.e., none of the above) | 295 |
| 1 | Transfer learning, meta learning, and lifelong learning | 274 |
| 2 | Datasets and benchmarks | 425 |
| 3 | Representation learning for computer vision, audio, language, and other modalities | 360 |
| 4 | Unsupervised, self-supervised, semi-supervised, and supervised representation learning | 686 |
| 5 | Generative models | 844 |
| 6 | Reinforcement learning | 637 |
| 7 | Applications to physical sciences (physics, chemistry, biology, etc.) | 408 |
| 8 | Applications to neuroscience & cognitive science | 166 |
| 9 | Learning theory | 284 |
| 10 | Causal reasoning | 96 |
| 11 | Neurosymbolic & hybrid AI systems (physics-informed, logic & formal reasoning, etc.) | 95 |
| 12 | Probabilistic methods (Bayesian methods, variational inference, sampling, UQ, etc.) | 220 |
| 13 | Applications to robotics, autonomy, planning | 196 |
| 14 | Learning on graphs and other geometries & topologies | 320 |
| 15 | Societal considerations including fairness, safety, privacy | 214 |
| 16 | Optimization | 328 |
| 17 | Visualization or interpretation of learned representations | 87 |
| 18 | Metric learning, kernel learning, and sparse coding | 6 |
| 19 | Infrastructure, software libraries, hardware, systems, etc. | 50 |
| 20 | Applications to computer vision, audio, language, and other modalities | 574 |
| 21 | Alignment, fairness, safety, privacy, and societal considerations | 499 |
| 22 | Interpretability and explainable AI | 249 |
| 23 | Foundation or frontier models, including LLMs | 670 |
| 24 | Learning on time series and dynamical systems | 116 |
| 25 | Other topics in machine learning (i.e., none of the above) | 223 |

The top-left $n \times n$ block equals $\mathrm{Diag}(z_1, \ldots, z_n) - M^+ B$, whose determinant is a nonzero polynomial (the $z_1 \cdots z_n$ coefficient equals 1), hence it is invertible for almost all $z_c$; thus $D\varphi_o(z_c)$ is invertible generically. There exists (up to scale) a unique nonzero $w$ with

$$
w^\top \left( \begin{pmatrix} z_1 & 0 & \cdots & 0 \\ & \ddots & \ddots & \vdots \\ 0 & \cdots & z_n & 0 \\ 0 & \cdots & 0 & 0 \end{pmatrix} - \tilde{M}^+ B \right) = 0, \qquad w^\top \left( \begin{pmatrix} \tilde{z}_1 & 0 & \cdots & 0 \\ & \ddots & \ddots & \vdots \\ 0 & \cdots & \tilde{z}_n & 0 \\ 0 & \cdots & 0 & 0 \end{pmatrix} - \tilde{M}^+ B \right) = 0.
$$

(34)

If some $w_i$ vanished on a set of positive measure, either the upper block would lose rank (contradicting the generic invertibility) or a minor involving the bottom row would force $(v^\top B)_1 = 0$, violating Assumption ii. Hence $w_i \neq 0$ for all $i \leq n$ almost everywhere, and subtracting the two displays gives

$$
(w_1(z_1 - \tilde{z}_1), \ldots, w_n(z_n - \tilde{z}_n), 0) = 0,
$$

(35)

so $z_i = \tilde{z}_i$ for all $1 \leq i \leq n$. Therefore $\varphi_o(z_c) = z_c$ almost everywhere (and by continuity everywhere), which implies

$$
\langle e_i, \tilde{g}^{-1}(x) \rangle = \langle e_i, g^{-1}(x) \rangle, \qquad 1 \leq i \leq n.
$$

(36)

Thus the concept coordinates, hence the true concepts $\mathbf{c} = A\mathbf{z}$, are identified up to permutation and coordinate-wise scaling. Since $\tilde{\mathbf{c}}$ enters only as weak supervision with independent Gaussian noise, it does not alter the identifiability class of $\mathbf{c}$; rather, it guides estimation.

This completes the proof of Theorem 1.

---

**Prompt 1.** *(Complete Prompt for Generating Concepts for Authors/Reviewers (1/2))*

*Task Description*
*You are an expert analyst tasked with evaluating an author-reviewer discussion from the OpenReview system. Your goal is to label the discussion content across 10 specific dimensions, as defined below, based on the provided reviewer comments, author rebuttals, and any additional interactions. For each dimension, assign a score (1, 2, 3, 4, or 5) according to the calibration criteria provided, and provide a brief justification (1-2 sentences) explaining your reasoning. This rubric is calibrated for top-tier machine learning conferences. A score of 3 reflects the expected standards from competent researchers at a top conference. Scores of 1–2 indicate responses that fall short of this benchmark, while 4–5 are reserved for rebuttals that are truly exceptional. Don't hesitate to give a 3 for strong, competent responses; use higher scores only for standout cases. Ensure your analysis is objective, precise, and grounded in the content.*

*Input Content:*
*The discussion content is provided in JSON format, containing reviewer summaries, strengths, weaknesses, questions, ratings, and author responses. Here is the content with one reviewer's comments to analyze:*

`{discussion}`

*Dimensions and Calibration Criteria:*

*1. Clarity of the Rebuttal (Presentation)*
*Definition: The extent to which the rebuttal communicates the authors' arguments and clarifications clearly, with logical structure, precise language, and proper grammar.*
*Calibration:*

- *1 (Weak): Generally understandable but with notable ambiguity or unclear phrasing.*
- *2 (Acceptable): Mostly clear, but with occasional awkwardness or minor lapses in flow or precision.*
- *3 (Competent): Well-structured and precise, meeting top-tier expectations.*
- *4 (Strong): Exceptionally clear and engaging, with polished language and logical flow.*
- *5 (Exceptional): Exemplary clarity using outstanding prose or innovative formatting to enhance understanding.*

*2. Directness in Addressing Reviewer Concerns (Presentation)*
*Definition: The degree to which the rebuttal directly and comprehensively responds to the reviewer's specific criticisms and questions.*
*Calibration:*

- *1 (Inadequate): Fails to meaningfully engage with reviewer concerns.*
- *2 (Partially Direct): Responds to some points but omits or glosses over others.*
- *3 (Fully Direct): Addresses all major concerns clearly and completely.*
- *4 (Very Direct): Thoughtful, complete, and anticipates follow-ups.*
- *5 (Exceptionally Direct): Insightful, persuasive, and goes beyond expectations in addressing concerns.*

*3. Positive Attitude (Presentation)*
*Definition: The extent to which the rebuttal maintains a constructive, respectful, and collaborative tone, even when disagreeing with reviewers.*
*Calibration:*

- *1 (Negative): Dismissive or combative tone.*
- *2 (Slightly Defensive): Polite but subtly frustrated or curt.*
- *3 (Constructive): Respectful, professional, and collaborative.*
- *4 (Gracious): Appreciative and collegial, fostering a positive tone.*
- *5 (Diplomatic): Extremely professional and generous in tone, even under criticism.*

*4. Willingness to Acknowledge Limitations or Propose Changes (Presentation)*
*Definition: The authors' openness to revising their work and candidly acknowledging limitations raised by reviewers.*
*Calibration:*

- *1 (Resistant): Avoids or dismisses valid concerns.*
- *2 (Minimally Open): Acknowledges minor issues with superficial fixes.*
- *3 (Open): Candidly acknowledges limitations and proposes improvements.*
- *4 (Receptive): Actively proposes meaningful adjustments.*
- *5 (Reflective): Embraces feedback and suggests substantial changes with humility.*

*5. Strength of Evidence or Justification (Technical)*
*Definition: The robustness of the rebuttal's claims, supported by experiments, references, logical reasoning, or other concrete evidence.*
*Calibration:*

- *1 (Weak): Vague or unsupported claims.*
- *2 (Partially Supported): Limited or insufficient justification.*
- *3 (Well Justified): Solid and relevant evidence or logic.*
- *4 (Thorough): Multiple, well-integrated forms of support.*
- *5 (Persuasive): Deep, compelling evidence demonstrating technical mastery.*

*6. Technical Convincingness and Rigor (Technical)*
*Definition: The technical soundness, rigor, and depth of understanding demonstrated in the rebuttal's arguments.*
*Calibration:*

- *1 (Unconvincing): Contains technical errors or vague reasoning.*
- *2 (Partially Convincing): Shows some understanding but includes logical gaps.*
- *3 (Solid and Sound): Technically correct and well-reasoned.*
- *4 (Insightful): Demonstrates thoughtful, deeper understanding.*
- *5 (Exceptional): Reveals technical mastery and strengthens the paper's core claims.*

---

**Prompt 2.** *(Complete Prompt for Generating Concepts for Authors/Reviewers (2/2))*

---

*7. Handling of Misunderstandings and De-escalation*
*Definition: The extent to which the authors recognize and address misunderstandings in reviewer comments while maintaining a constructive tone.*
*Calibration:*

- *1 (Defensive): Escalates tension with sarcasm or dismissiveness.*
- *2 (Tense): Curt or irritated tone; misunderstanding remains partially unresolved.*
- *3 (Professional): Clarifies misunderstandings calmly and clearly.*
- *4 (Tactful): Resolves issues gracefully and respectfully.*
- *5 (Masterful): Turns conflict into constructive dialogue with diplomacy.*

*8. Review Specificity and Constructiveness*
*Definition: The specificity, actionability, and constructiveness of the reviewer's feedback provided to the authors.*
*Calibration:*

- *1 (Vague): Lacks actionable suggestions.*
- *2 (Somewhat Specific): Contains some helpful points but is mixed with generalities.*
- *3 (Constructive): Clear, specific, and actionable feedback.*
- *4 (Thorough): Covers various aspects with a balanced and helpful tone.*
- *5 (Exemplary): Deeply engaged, precise, and improvement-oriented feedback.*

*9. Reviewer Open-mindedness*
*Definition: The reviewer's apparent willingness to reconsider their evaluation based on a compelling rebuttal.*
*Calibration:*

- *1 (Rigid): Unwilling to engage or revise stance.*
- *2 (Cautious): Skeptical, with minimal openness.*
- *3 (Reasonable): Acknowledges merit and shows willingness to revise.*
- *4 (Flexible): Thoughtfully re-evaluates if rebuttal is persuasive.*
- *5 (Proactive): Encourages rebuttal and signals readiness to change stance.*

*10. Severity of Concerns*
*Definition: The seriousness of the reviewer's criticisms, influencing the rebuttal's potential to change the reviewer's opinion.*
*Calibration:*

- *1 (Minor): Only minor or editorial comments.*
- *2 (Moderate): Substantive but addressable issues.*
- *3 (Serious): Challenges to key aspects of the paper.*
- *4 (Major): Foundational doubts about core contributions.*
- *5 (Critical): Calls the publishability of the work into question.*

**Instructions**

1. *Analyze the provided content, focusing on the author's rebuttal (if any) and the reviewer's comments.*
2. *For each of the 10 dimensions, assign a score (1, 2, 3, 4, or 5) based on the calibration criteria.*
3. *Provide a brief justification (1-2 sentences) for each score, referencing specific aspects of the content.*
4. **Summary:** *After analyzing all, provide a summary string containing the 10 dimension scores separated by hyphens.*
5. *If a dimension cannot be evaluated due to insufficient information (e.g., no rebuttal for technical dimensions), assign a score 0 and explain why.*
6. *If the content references another reviewer (e.g., "reviewer ciFG" or "reviewer sq8T") whose comments are not provided, note that the analysis is limited to the available content.*
7. *Ensure your tone remains neutral and professional, focusing on the content's quality and alignment with the criteria.*
8. *Present your output strictly in the following format:*
   *Output Format*
   1. **Clarity of the Rebuttal:** *[Score]*
      Justification: *[1–2 sentences]*
   2. **Directness in Addressing Reviewer Concerns:** *[Score]*
      Justification: *[1–2 sentences]*
   3. **Positive Attitude:** *[Score]*
      Justification: *[1–2 sentences]*
   4. **Willingness to Acknowledge Limitations or Propose Changes:** *[Score]*
      Justification: *[1–2 sentences]*
   5. **Strength of Evidence or Justification:** *[Score]*
      Justification: *[1–2 sentences]*
   6. **Technical Convincingness and Rigor:** *[Score]*
      Justification: *[1–2 sentences]*
   7. **Handling of Misunderstandings and De-escalation:** *[Score]*
      Justification: *[1–2 sentences]*
   8. **Review Specificity and Constructiveness:** *[Score]*
      Justification: *[1–2 sentences]*
   9. **Reviewer Open-mindedness:** *[Score]*
      Justification: *[1–2 sentences]*
   10. **Severity of Concerns:** *[Score]*
       Justification: *[1–2 sentences]*
   
   ***Summary:*** *[Score1–Score2–...–Score10].*

Figure A3: **Conditional Independence Test**. The p-value of KCI (blue) and RCIT (orange).

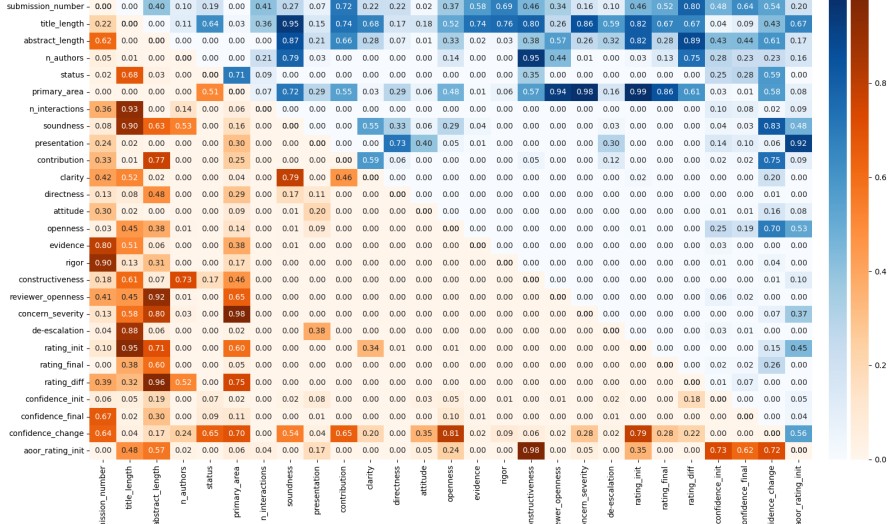

Figure A4: **Conditional Independence Test**. The p-value of HSIC (blue) and Chi-square (orange).

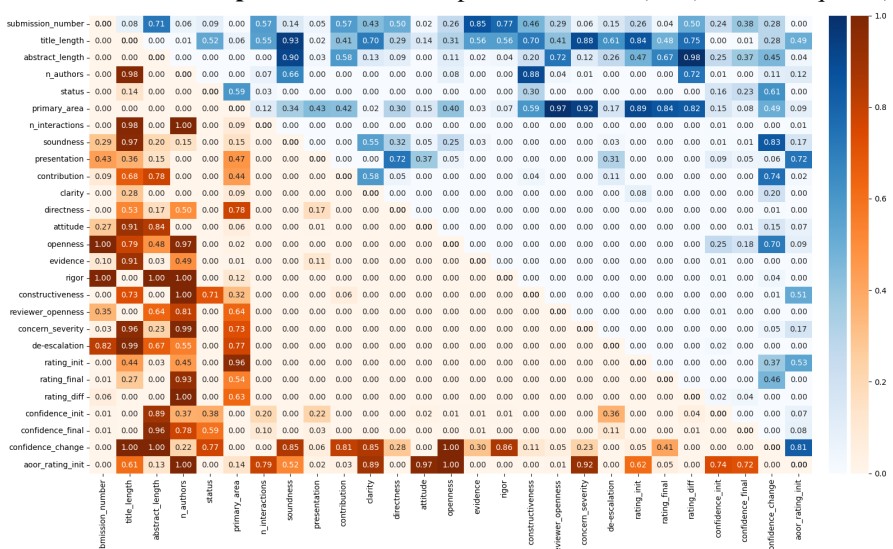

Figure A5: **Conditional Independence Test**. The p-value of GSQ (orange).

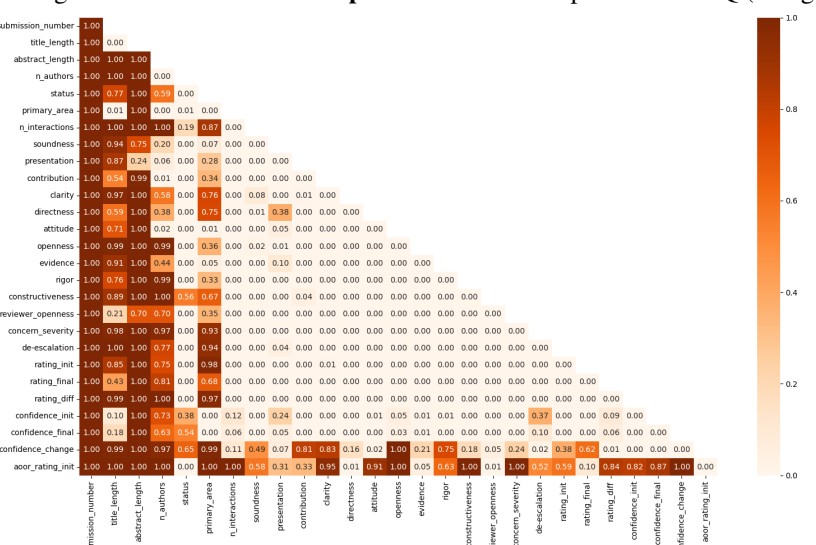

Figure A6: **Dependency Panels** of rating_init

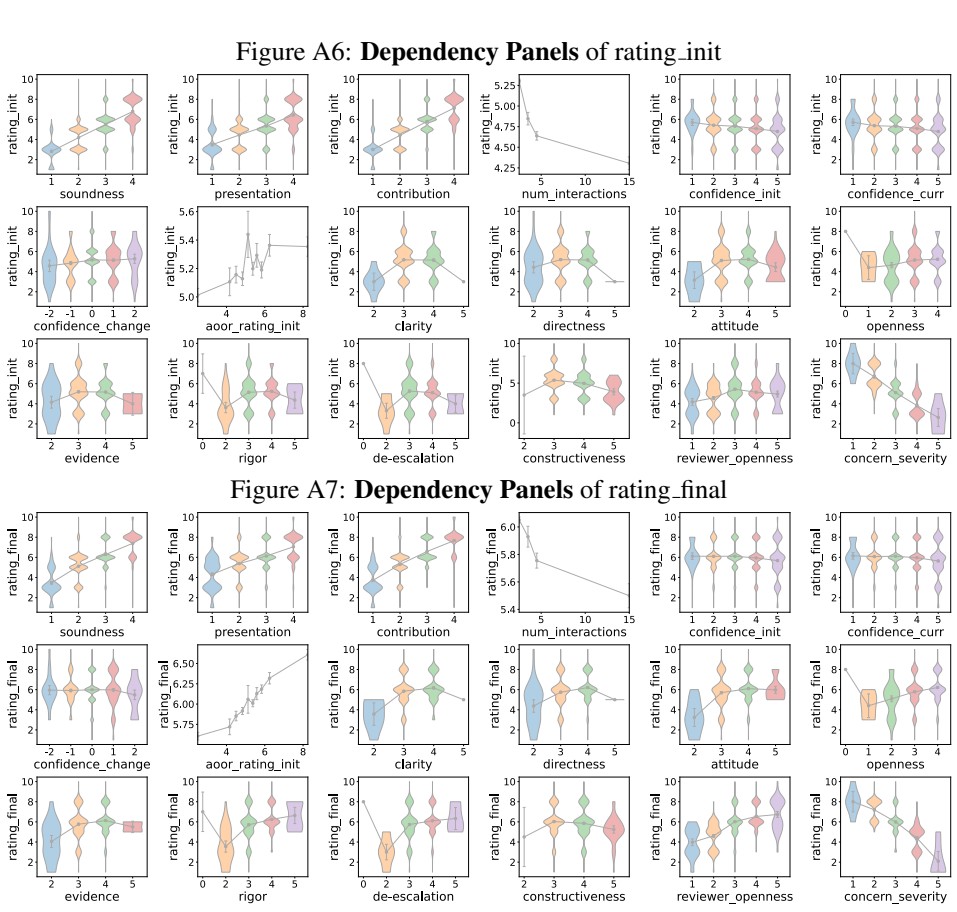

Figure A7: **Dependency Panels** of rating_final

