# OpenReview forum: "How Effective is Your Rebuttal? Identifying Causal Models from the OpenReview System"
_ICLR.cc/2026/Conference — ICLR 2026 Conference Withdrawn Submission_

### Official Review · Reviewer_2xmX · 2025-10-27

**Soundness:** 2
**Presentation:** 3
**Contribution:** 2
**Rating:** 4
**Confidence:** 4

**Summary:**

The paper investigates how author rebuttals influence final review decisions by conducting a two-layer causal analysis of ICLR submissions. At the structured level, it combines metadata and LLM-inferred features to test their associations with rating changes. At the unstructured level, it applies a weakly supervised Causal Representation Learning (CRL) framework on rebuttal texts, using concept-level supervision from LLM-inferred features. The study provides theoretical identifiability guarantees and empirically reveals how specific rebuttal strategies shape reviewer assessments, offering actionable guidance for crafting more effective rebuttals.

**Strengths:**

- The authors propose a two-layer causal analysis framework (structured + unstructured) leveraging real-world ICLR OpenReview data, and develop a weakly supervised Causal Representation Learning (CRL) model for rebuttal texts with concept-level supervision.

- The integration of the two levels is well-motivated: the CRL builds directly on the structured analysis, refining noisy concept features while also discovering new latent concepts. This design enables a richer understanding of how rebuttal strategies shape reviewer assessments.

- The paper provides both theoretical identifiability guarantees (Identifiability of Concepts) and empirical insights into effective rebuttal strategies.

- The visualizations, especially the violin plots, effectively and intuitively illustrate the relationships between rating changes and various features. Some conclusions — such as “more rebuttal rounds tend to increase scores,” “rating decreases often correspond to increased confidence,” and “higher clarity/directness tends to yield higher score improvements” — align well with human intuition.

**Weaknesses:**

- The experiments are conducted only on 8,000+ ICLR 2024–2025 papers, which raises concerns about the scale and diversity of the dataset. It remains unclear whether this is sufficient for robust generalization.

- The authors employ 8 review-level, 6 paper-level, and 10 LLM-inferred features, meaning that nearly half of the features (10/24) are LLM-generated. However, the reliability and consistency of these LLM-inferred features — and how they affect downstream analysis — are not clearly justified.

- The study focuses solely on textual characteristics of papers and rebuttals, without considering non-textual (e.g., visual or figure-related) information, which limits the comprehensiveness of the analysis.

- The problem setting is restricted to examining the effect of rebuttal behavior on final rating changes, without accounting for the underlying paper quality. For instance, poor-quality submissions may not benefit from clear and well-written rebuttals. Such latent factors could potentially confound the causal conclusions drawn.

**Questions:**

1. Why are the experiments limited to the 8,000+ papers from ICLR 2024–2025? While ICLR’s open-review transparency is understandable, why not include data from additional years to enhance robustness?

2. For the LLM-inferred features, are the detailed generation settings (e.g., prompts) provided? How is the reliability of these AI-generated features ensured to support subsequent analyses?

3. In Figure 3, why does concern_severity show an increase–then–decrease trend with rating_change? Does the paper provide a reasonable explanation? Could this pattern instead arise from the instability of the LLM-generated metric itself?

4. I noticed the existence of a related dataset, Re2 [1], which includes multi-year, multi-conference peer review and rebuttal data. Could the authors extend their experiments to Re2 to test whether the conclusions hold across broader settings? Such an extension would significantly strengthen the paper’s robustness.

5. The lack of multimodal analysis (e.g., incorporating figures or visual data) should be explicitly discussed as a limitation in the paper.

6. In generating concept scores using the prompt “Don’t hesitate to give a 3 for strong, competent responses; use higher scores only for standout cases,” why did the authors adopt this conservative scoring encouragement? Wouldn’t a wider score distribution (greater discrimination) be preferable?

> [1] Re2: A Consistency-ensured Dataset for Full-stage Peer Review and Multi-turn Rebuttal Discussions.

---

### Official Review · Reviewer_Aec3 · 2025-10-31

**Soundness:** 2
**Presentation:** 2
**Contribution:** 2
**Rating:** 2
**Confidence:** 3

**Summary:**

This paper investigates what makes rebuttals effective in changing reviewer ratings, analyzing ICLR 2024-2025 submissions from OpenReview. The authors perform a two-layer analysis. First, they apply conditional independence tests to structured metadata features and 10 LLM-inferred concepts derived from rebuttal and review texts (7 and 3 resp). Second, they apply weakly supervised Causal Representation Learning (CRL) directly to review and rebuttal text, using the noisy LLM-inferred concepts as weak supervision.

**Strengths:**

The peer review process is an important real-world setting, and this paper seeks to draw conclusions about the influencing factors based on analysis of a real-world dataset. The authors do perform some original analysis of the dataset and testing of different causal methods---though fail to demonstrate the generality of their findings. The paper is largely clear in its writing, though leaves out key details.

**Weaknesses:**

- In Section 3, the authors make conclusions about the "drivers of reviewer updates" that are inappropriate based on independence tests alone -- those factors like "persuasiveness, clarity, and responsiveness" may be correlated rather than drivers.
- Section 3 is also missing contextualization of how their results contribute to current knowledge (related work).
- The introduction provides a paragraph on prior work, which is insufficient and refers to the appendix rather than elaborating in the body text as is standard.
- "We aggregate each paper’s reviews without accounting for the precise timestamps of individual revisions." - this seems like a dangerous assumption. It's crucial to ensure that predictions are conditioned only on past events when drawing conclusions about the driving factors, as is the goal of this paper.
- The CRL analysis does not meaningfully improve the trustworthiness of the conclusions. Please see the questions about the synthetic setup, and conclusions drawn from the brief real-world section (4.5).

**Questions:**

- Why use Paper Copilot and not the ICLR data directly? How reliable is that website?
- How are the independence tests appropriate for the features, or what processing was done to make them appropriate? The data appears ordinal or continuous which does not meet the requirements for all the tests (eg chi-square).
- "with disagreements resolved through discussion to produce gold-standard scores" (p. 3): What was the rate of disagreements?
- "Interestingly, rating init and rating final appear independent of each other but both are dependent on rating change, suggesting that absolute ratings and their shifts capture complementary aspects of the review process." This doesn't seem supported by Fig 1 - please explain.
- Why is the synthetic data generated in Section 4.4 appropriate?
- "the Author Unknown node, connected to openness and influencing reviewer openness, likely reflects hidden aspects of persuasiveness or tone in author responses, while the Reviewer Unknown node, linked with review quality and concern severity and directly affecting rating change, appears to
capture latent reviewer dispositions such as strictness or flexibility" - what additional evidence do the authors have to back these guesses about the meaning of these variables? Have the authors performed any analyses?

---

### Official Review · Reviewer_GsGe · 2025-10-31

**Soundness:** 3
**Presentation:** 3
**Contribution:** 2
**Rating:** 4
**Confidence:** 2

**Summary:**

The paper combines dependence testing on structured/LLM-inferred features with a CRL model to uncover which rebuttal strategies causally relate to rating changes in OpenReview.

They conduct a two-layer analysis linking discourse quality features (clarity, evidence, rigor, openness, etc.) and interaction volume to rating changes, showing weak effects from static paper metadata.

They introduce a CRL framework with identifiability guarantees to recover human-aligned latent concepts from rebuttal/review text.

The learned causal graph highlights how author clarity/openness and reviewer openness/severity connect to rating changes, and it surfaces additional latent factors.

**Strengths:**

Understanding what moves reviewer scores during rebuttals is valuable for authors, reviewers, and program committees. This is especially interesting as an ICLR 26 submission author!

The authors provide explicit disclosure of the LLM labeling setup, model selection (DeepSeek-R1), and permissions for dataset usage.

**Weaknesses:**

The paper’s overall framing and emphasis (rebuttal effectiveness) make it seem like a meta-study rather than a core ML contribution. The causal layer feels incremental, not groundbreaking. It’s well-motivated and technically solid, but not strong enough on its own to carry an ICLR acceptance. This paper might do better in an applied ML venue or a technical blogpost track.

Aggregating the thread loses who responded when and what changed after which message. This could confound cause/effect timing. Even coarse timestamps or a panel model (pre/post rebuttal message windows) would strengthen the narrative.

**Questions:**

Some other questions I think would be interesting to answer :

1. How does reviewer confidence (both reported and gauged confidence) affect score change? Are highly confident reviewers more likely or unlikely to change scores? What confounding variables influence this decision?

2. Can the LLMs predict the time invested by the reviewer in reading a paper and writing the review (i.e shallowness of the review). How does the shallowness of the review affect the chance of a rebuttal score change?

3. What happens where there is a lot of variance in reviewer scores? Are reviewers incentvized to align with each other?

4.  Did you identify any way to game rebuttals? For example, some authors present a summary of all rebuttal discourse for the meta reviewer to review at a glance. Some authors are overly grateful, some overly direct.  Are these influential in the final paper decision?

---

### Official Review · Reviewer_6XKW · 2025-10-31

**Soundness:** 3
**Presentation:** 4
**Contribution:** 1
**Rating:** 2
**Confidence:** 4

**Summary:**

Provide a two-layer causal analysis of peer reviews, studying authors' influence to final decisions.

**Strengths:**

Did in-depth analysis of peer reviews on metadata, reviewl-level features, and LLM-inferred concepts.
The design and formulation of the second-layer causal framework seems well executed and grounded.

**Weaknesses:**

Can't understand how the list of 10 LLM concepts is chosen. Following text, I also checked Appendix (e.g., A4.1), but still don't rationales behind it.
Most findings from review-level metadata features seems quite obvious, not giving additional excitement, while findings from LLM-inferred features are not reliable unless human verification performed.
I wish to learn more about the application of the trained model on real-world peer-review data and more in-depth discussion, but only visual diagram of inferred graph is given without details.
Also, the causal network is primarily designed by unverified concepts and their labels in synthetic data, and I don't know how much I should trust the output from the CRL model.

**Questions:**

See my comments above

---

### Author Response · Authors · 2025-11-21

We sincerely thank all reviewers for their thoughtful and constructive feedback. Your comments on the methodology, conceptual framing, and evaluation have been invaluable in helping us better understand the limitations and future potential of this work. We will further refine the ideas and experiments based on your suggestions and hope to resubmit an improved version in the future. Thank you again for your time and detailed engagement with our submission!

---

### Note · Authors · 2025-11-21

I have read and agree with the venue's withdrawal policy on behalf of myself and my co-authors.